# Clinical Implications of Neurological Comorbidities and Complications in ICU Patients with COVID-19

**DOI:** 10.3390/jcm10112281

**Published:** 2021-05-25

**Authors:** Jaeseok Park, Yong-Shik Kwon, Hyun-Ah Kim, Doo-Hyuk Kwon, Jihye Hwang, Seong-Hwa Jang, Hyungjong Park, Sung-Il Sohn, Huimahn Alex Choi, Jeong-Ho Hong

**Affiliations:** 1Department of Internal Medicine, Division of Pulmonology, Keimyung University Dongsan Hospital, Keimyung University School of Medicine, Daegu 42601, Korea; mdpjs79@naver.com (J.P.); kwonys0515@naver.com (Y.-S.K.); 2Division of Infectious Disease, Department of Internal Medicine, Keimyung University Dongsan Hospital, Keimyung University School of Medicine, Daegu 42601, Korea; hyunah1118@dsmc.or.kr; 3Department of Neurology, Daegu Dongsan Hospital, Keimyung University School of Medicine, Daegu 41931, Korea; kspy1d@naver.com (D.-H.K.); jhhwang0110@gmail.com (J.H.); 4Department of Neurology, Yeungnam University Hospital, Yeungnam University School of Medicine, Daegu 42415, Korea; 5Department of Neurology, Keimyung University Dongsan Hospital, Keimyung University School of Medicine, Daegu 42601, Korea; seonghwajang0416@gmail.com (S.-H.J.); hjpark209042@gmail.com (H.P.); sungil.sohn@gmail.com (S.-I.S.); 6Department of Neurology and Neurosurgery, McGovern Medical School, University of Texas Health Science Center, Houston, TX 77030, USA; Huimahn.A.Choi@uth.tmc.edu

**Keywords:** coronavirus, neurology, critical care, neurocritical care, mortality

## Abstract

Clinical implications of neurological problems during intensive care unit (ICU) care for coronavirus disease 2019 (COVID-19) patients are unknown. This study aimed to describe the clinical implications of preexisting neurological comorbidities and new-onset neurological complications in ICU patients with COVID-19. ICU patients who were isolated and treated for COVID-19 between 19 February 2020 and 3 May 2020, from one tertiary hospital and one government-designated branch hospital were included. Clinical data including previous neurological disorders were extracted from electronic medical records. All neurological complications were evaluated by neurointensivists. Multiple logistic regression analysis was performed to investigate independent factors in ICU mortality. The median age of 52 ICU patients with COVID-19 was 73 years. Nineteen (36.5%) patients had preexisting neurological comorbidities, and new-onset neurological complications occurred in 23 (44.2%) during ICU admission. Patients with preexisting neurological comorbidities required tracheostomy more frequently and more ventilator and ICU days than those without. Patients with new-onset neurological complications experienced more medical complications and had higher ICU severity score and ICU mortality rates. New-onset neurological complications remained an independent factor for ICU mortality. Many COVID-19 patients in the ICU have preexisting neurological comorbidities, making them at a high risk of new-onset neurological complications.

## 1. Introduction

The novel coronavirus disease (COVID-19) outbreak in Wuhan, China, has resulted in an ongoing pandemic and is associated with high morbidity and mortality, particularly in conjunction with old age and some underlying medical conditions [1,2,3]. Recent reports from the Centers for Disease Control and Prevention (CDC) in the United States have indicated that cancer, chronic kidney disease, heart failure, coronary artery disease, and type 2 diabetes mellitus, for example, put patients at an increased risk for severe illness from COVID-19 [4]. However, the clinical effects of underlying neurological conditions remain uncertain because of a lack of evidence. 

COVID-19 can lead to various neurological complications and lethal respiratory infections such as acute respiratory distress syndrome (ARDS) [1]. Neurological manifestations were found in 36.4% of COVID-19 patients admitted to the hospital in Wuhan, China, which were more often found in more severe disease [5]. Serious neurological complications, including acute stroke and impaired consciousness, were relatively common [5,6]. In addition, data from the two intensive care units (ICUs) for COVID-19 patients in France showed that agitation and delirium were observed in approximately two-thirds of ICU patients, and neurological findings were observed in 67% of patients when sedation and neuromuscular blockers were discontinued [7]. However, little is known about how new-onset neurological complications affect severe COVID-19 patients.

Herein, we aimed to investigate the clinical implications of preexisting neurological comorbidities and new-onset neurological complications in COVID-19 patients admitted to the ICU.

## 2. Materials and Methods

### 2.1. Selection of Study Subjects

The city of Daegu emerged as the epicenter of COVID-19 in South Korea and has accounted for more than two-thirds of the over 10,000 infections in the country. A total of 1004 patients were isolated and treated for COVID-19 from 19 February 2020 to 3 May 2020, in two hospitals in Daegu: Keimyung University Dongsan Hospital (a tertiary hospital) and Daegu Dongsan Hospital (a government-designated branch hospital). The government-designated branch hospital served as a regional hub to exclusively treat COVID-19 patients. As the number of new cases of COVID-19 decreased rapidly in Daegu, the ICU of the branch hospital was closed on 3 May 2020, which allowed us to obtain the final medical outcomes from most of the patients.

All subjects were tested for severe acute respiratory syndrome coronavirus 2 (SARS-CoV-2) using a real-time reverse transcription–polymerase chain reaction (RT–PCR) assay of a nasal swab specimen based on the recommendations of the CDC and the Korean CDC [8,9]. Patients who were admitted to the integrated ICU for critical care were enrolled in our study. A collaborative approach was designed and implemented by our critical care team, and various subspecialized intensivists were dispatched by the Korean Society of Critical Care Medicine.

### 2.2. Ethical Considerations

Our study was approved by the local institutional review boards (IRBs; IRB No. 2020-05-048) of the hospitals for the collection of anonymized clinical data with a waiver of informed consent because of the study’s retrospective design, subject anonymity, and minimal risk to participants. Patients’ clinical characteristics, laboratory parameters, treatments, and outcomes were obtained from the Dongsan COVID-19 registry. This study was conducted in accordance with the Declaration of Helsinki.

### 2.3. Data Collection and Definitions

The criteria for ICU admission included clinical conditions in which intensive care and continuous monitoring were required because of severe symptoms accompanied by multiple organ failure after a positive COVID-19 diagnosis. Specific conditions included severe pneumonia (respiration rate ˃30 or ˂90% oxygen saturation with dyspnea), ARDS, and sepsis (with or without shock).

Specific signs and symptoms of COVID-19 were collected from questionnaires taken at the time of hospitalization and included fever, chills, cough, sputum production, rhinorrhea, sore throat, myalgia or fatigue, headache, diarrhea, dyspnea, and chest pain. At the early period of the COVID-19 crisis, new-onset loss of taste or smell was not a well-known symptom; therefore, it was not assessed in all ICU patients.

Preexisting neurological comorbidities were ascertained through self-reporting, medication review, and hospital record review. Preexisting neurological comorbidities were divided into (1) neurodegenerative diseases, (2) cerebrovascular diseases, and (3) other neurological diseases. Neurodegenerative diseases included Alzheimer’s disease and other dementias, Parkinson’s disease (PD) and PD-related disorders, and motor neuron diseases. Cerebrovascular diseases were defined as any disorder in which an area of the brain was temporarily or permanently affected by ischemia or bleeding. The categories in “other neurological diseases” included all other neurological disorders except neurodegenerative and cerebrovascular diseases. 

Clinical complications included new-onset neurological complications, ARDS, septic shock, and acute kidney injury (AKI). All these complications that occurred from the ICU admission day were collected. New-onset neurological complications were seizure, acute stroke, diffuse hypoxic brain injury, and delirium. Seizure was diagnosed as a clinical seizure or electrical seizure activity in the electroencephalogram (EEG) without any clinical signs of seizure (non-convulsive seizure) if COVID-19 patients had unexplained mental changes. Acute stroke was defined as an acute neurological deficit with focal signs and symptoms upon neurological examination, regardless of symptom duration, and was classified as ischemic or hemorrhagic based on computed tomography (CT) or magnetic resonance imaging (MRI). Delirium was diagnosed by intensivists using the Intensive Care Delirium Screening Checklist or Confusion Assessment Method for the ICU. All new-onset neurological complications during ICU treatment were evaluated and confirmed by neurointensivists. Seizure and acute stroke were treated by each guideline. Principles of delirium management were supportive medical care and treatment of underlying conditions. If indicated, antipsychotics such as quetiapine or haloperidol were administered for severe agitation [10].

The definition of ARDS was based on the World Health Organization (WHO) guidelines for COVID-19 [11]. Briefly, ARDS was defined as a PaO_2_/FiO_2_ ratio <300 mmHg with a positive end-expiratory pressure ≥5 cm H_2_O within 1 week of known clinical insult, or new or worsening respiratory symptoms. Septic shock was defined as persistent hypotension despite volume resuscitation requiring vasopressors to maintain a mean arterial pressure ≥65 mmHg and a serum lactate level >2 mmol/L. AKI was defined using the Kidney Disease Improving Global Outcome criteria [12]. AKI was diagnosed if any one of the following was present: (1) a serum creatinine increase of ≥0.3 mg/dl within 48 h, (2) a serum creatinine increase of ≥1.5 times the baseline within 7 days, or (3) a urine volume <0.5 mL/kg/h for 6 h. 

The Sequential Organ Failure Assessment (SOFA) score reflects organ failure/dysfunction over time, which can predict mortality in critically ill patients [13]. We used the patients’ worst recorded values (PaO_2_/FiO_2_ ratio, platelet count, bilirubin, mean arterial pressure with inotropes, Glasgow Coma Score, and creatinine or urine output) within 24 h of ICU admission to calculate SOFA scores and evaluate baseline severity. Acute Physiology and Chronic Health Evaluation (APACHE) II score within 24 h of ICU admission with worst value recorded for each component part of 12 physiology variables was also employed [14].

Time variables were calculated with respect to length of ICU stay, ventilator days, and symptom onset-to-ICU admission. If a patient no longer required critical care and intensive monitoring in the ICU, as determined by intensivists, they were discharged or transferred from the ICU regardless of the RT–PCR results. The rate of transfer out of the ICU and ICU death were reported as clinical outcomes.

### 2.4. Statistical Analysis

Categorical variables are presented as counts and percentages and continuous variables as median and interquartile range (IQR) because of the small number of subjects. To compare the baseline characteristics according to the existence of neurological comorbidities or complications, the Pearson χ^2^-test or Fisher’s exact test was used for categorical variables and Mann–Whitney U test was used for continuous variables, because of the number of subjects assigned to each group. Significance levels were set at *p* < 0.05 for two-tailed tests. Cumulative risks of new-onset neurological complications in the two groups according to preexisting neurological comorbidities were calculated using Kaplan–Meier estimates. The Cox proportional hazard model was applied to evaluate the influences on new-onset neurological complications according to preexisting neurological comorbidities. Multiple logistic regression analysis was used to evaluate the independent predictors of ICU mortality in ICU patients with COVID-19. Predefined variables (age, sex, and APACHE II score) were adjusted in both of the statistical analysis models because of an overfitting issue. All statistical analyses were performed using SPSS (version 19.0; SPSS Inc., Chicago, IL, USA).

## 3. Results

Of the 1004 patients infected with COVID-19 admitted to the integrated ICU for critical care from the two hospitals, 52 were included in the study, including 15 of 40 patients from the tertiary hospital and 37 of 964 from the government-designated branch hospital. The median age of our patient cohort was 73 (IQR 57–81) years, and 51.9% were male. Patients with preexisting neurological comorbidities (79 (59–84) vs. 68 (56–78) years, *p* = 0.114) or new-onset neurological complications (79 (63–85) vs. 68 (54–77) years, *p* = 0.054) tended to be old. The proportion of male patients with preexisting neurological comorbidities was higher than those without (73.7% vs. 39.4%, *p* = 0.017). At baseline, no significant differences were found in the proportion of underlying medical conditions between the preexisting neurological comorbidity and new-onset neurological complication groups. The median APACHE II and SOFA scores at admission were 11 (7–21) and 4 (2–6), respectively. Although no significant difference was observed in the APACHE II and SOFA score between the two groups according to preexisting neurological comorbidities (12 (9–21) vs. 10 (7–20), *p* = 0.232; 4 (2–7) vs. 3 (2–5), *p* = 0.343), patients with new-onset neurological complications had significantly higher APACHE II and SOFA scores than those without (12 (9–22) vs. 9 (7–19), *p* = 0.040; 5 (2–7) vs. 2 (1–5), *p* = 0.044). Among 52 ICU patients, 25% did not complain of COVID-19-related signs or symptoms at initial admission. Patients with preexisting neurological comorbidities had significantly lower rates of sore throat (0% vs. 27.3%, *p* = 0.018), myalgia or fatigue (10.5% vs. 54.5%, *p* = 0.002), and headache (5.3% vs. 33.3%, *p* = 0.037) than those without (Table 1).

Table 2 shows preexisting neurological comorbidities and new-onset neurological and medical complications. Preexisting neurological comorbidities existed in 19 (36.5%) patients, including seven patients with neurodegenerative diseases (four with Alzheimer’s disease or other dementias, two with PD and PD-related disorders, and two with motor neuron disease), seven patients with cerebrovascular diseases, and five patients with other neurological diseases, all of which were present before ICU admission. Two preexisting neurological comorbidities overlapped in one patient (PD with dementia). The five cases of other neurological diseases included one case of spasmodic dystonia, one case of encephalitis, two cases of epilepsy, and one case of traumatic brain injury.

As regards medical complications, ARDS, AKI, and septic shock occurred in 40.4%, 25%, and 42.3% of the patients, respectively. A total of 23 (44.2%) patients had new-onset neurological complications, three (5.8%) had acute stroke, three (5.8%) experienced seizures, and 20 (38.5%) had delirium. The median onset time of new-onset neurological complications was 8 (2–10) days after ICU admission: seizure, 8 (10–13) days; acute stroke, 11 (1–19) days; and delirium 2 (2–8) days. The three cases of acute stroke were all classified as ischemic. ARDS (60.9% vs. 24.2%, *p* = 0.017) and septic shock (60.9% vs. 27.6%, *p* = 0.033) occurred more frequently in patients with new-onset neurological complications than in those without.

Moreover, 50% of patients were treated with a mechanical ventilator; extracorporeal membrane oxygenation (ECMO) and continuous renal replacement therapy (CRRT) were applied in 9.6% and 15.4%, respectively. Patients with preexisting neurological comorbidities underwent tracheostomy more frequently (36.8% vs. 6.1%, *p* = 0.008). The durations (days) of invasive ventilation (23 vs. 5, *p* = 0.013) and ICU stay (27 vs. 16, *p* = 0.048) were longer in patients with preexisting neurological comorbidities than in those without. Mechanical ventilation was performed more frequently on patients with new-onset neurological complications (69.6% vs. 34.5%, *p* = 0.026) (Table 3).

Of the 19 patients with preexisting neurological comorbidities, 14 (73.7%) had new-onset neurological complications, which was significantly higher than that in the group without neurological complications (73.7% vs. 27.3%, *p* = 0.001). Figure 1 shows the Kaplan–Meier curve of the cumulative risk of new-onset neurological complications according to preexisting neurological comorbidities. The Cox proportional hazards model showed that preexisting neurological comorbidities was an independent predictor of future new-onset neurological complications (hazard ratio [HR], 2.894; 95% confidence interval [CI], 1.194–7.011, *p* = 0.019).

HR, hazard ratio; CI, confidence interval. Among the 52 patients, the ICU mortality rate was 33.3%. Patients with new-onset neurological complications had a significantly higher rate of ICU mortality than those without (56.5% vs. 20.7%, *p* = 0.008) (Table 3). After adjusting for age, sex, and APACHE II score, new-onset neurological complications (odds ratio [OR], 6.18; 95% CI, 1.16–32.90, *p* = 0.03) and APACHE II score (OR, 1.20; 95% CI, 1.06–1.36, *p* = 0.003) remained independent predictors of ICU mortality (Table 4). No differences in the results were observed when the SOFA score was included in the analysis instead of the APACHE II score (Appendix A). Patients with delirium had a significantly higher rate of preexisting neurological comorbidities (65.0% vs. 18.8%, *p* = 0.001), chronic kidney disease (25.0% vs. 3.1%, *p* = 0.026), and ICU mortality than those without (55.0% vs. 25.0%, *p* = 0.029) (Appendix A).

## 4. Discussion

To the best of our knowledge, this is the first study on the clinical implications of preexisting neurological comorbidities and new-onset neurological complications in ICU patients infected with COVID-19.

First, we focused on the effect of COVID-19 on ICU patients with preexisting neurological disease. Recently, García-Azorín et al. from Spain evaluated the association between the prognosis and comorbid chronic neurological disorders in COVID-19 patients and found that preexisting chronic neurological disorders were independent predictors of mortality in hospitalized COVID-19 patients [15]. Another study from Wuhan, China, reported that COVID-19 patients with preexisting stroke were readily predisposed to death (mortality between patients with and without stroke, 45% vs. 9%, adjusted HR 1.73 [95% CI, 1.00–2.98]) [16]. These studies were based on all hospitalized COVID-19 patients. Our study only enrolled patients admitted to the ICU who, compared with the patients of the abovementioned referenced studies, had a more severe condition based on the median SOFA score (4 in our study vs. 1 in the Chinese study) and mortality rate (33% in our study vs. 22% in the Spanish study vs. 12% in the Chinese study). In our study, preexisting neurological comorbidities increased the duration of ventilator use, length of ICU stay, and need for a tracheostomy. Those with neurological conditions are at an increased risk of ICU-acquired and ventilator-associated pneumonia because of impaired airway reflexes, including difficulties in swallowing and coughing [17,18]. Patients with preexisting neurological comorbidities had a significantly lower rate of subjective COVID-19-related symptoms, such as sore throat, myalgia, fatigue, or headache before admission; however, it is likely that patients with preexisting neurological comorbidities were not capable of self-reporting such symptoms. Cough symptoms were also observed at a relatively low rate in patients who already had neurological disorders, including stroke, head trauma, PD, and motor neuron disease, all of which may have reduced the cough reflexes. Therefore, a thorough patient history, including preexisting neurological comorbidities, and adequate airway management in patients with preexisting neurological comorbidities are important when addressing COVID-19-related symptoms. However, this did not affect the mortality rate of ICU patients with COVID-19. This is because both groups were already in a severe condition when they were admitted to the ICU, and the multidisciplinary team, including neurointensivists, involved in all care had a positive effect on the mortality rate [19].

This study also examined the effects of new-onset neurological complications in ICU patients with COVID-19 in relation to the clinical effect, including ICU mortality. Cases of direct involvement of SARS-CoV-2 into the central or peripheral nervous system are very rare [20]. However, recent evidence has emerged regarding a post-viral autoimmune phenomenon including indirect involvement [21,22,23,24]. Mao et al., from China, recently reported that one-third of COVID-19 patients have neurological symptoms, which are more common (45.5%) in patients with severe infections. In our study, new-onset neurological complications occurred in approximately 44% of patients who were treated for COVID-19 in the ICU. Severe neurological complications, such as acute cerebrovascular disease and consciousness disturbance, occurred more frequently in the later stages of the disease [5]. In our study, seizure and acute stroke occurred within >7 days of ICU admission.

Seizure has not been widely reported in patients with COVID-19. SARS-CoV-2 infection may cause multiple organ failures, metabolic derangements, and hypoxia, which can lead to clinical or subclinical seizures [25]. Acute cerebrovascular disease is considered a major cause of seizure. Acute stroke is an emerging COVID-19 complication, with cohort studies reporting stroke in 2%–6% of patients, similar to the rate seen in our study (5.8%) [26]. The link between SARS-CoV-2 and cerebrovascular disease is an important mechanism for new-onset neurological complications. SARS-CoV-2-induced inflammatory responses result in coagulation and thrombin generation and are followed by thromboembolism and D-dimer accumulation that can influence the onset or worsening of stroke [27,28,29]. ARDS, acute cardiac injury, and septic shock have been reported to occur frequently in ICU patients with COVID-19 [30,31]. Neurological complications may be caused by COVID-19-related medical complications [7,32]. Delirium in the ICU has a negative effect on the outcome of patients [32,33].

Preexisting neurological comorbidities can affect new-onset neurological complications. Patients with neurodegenerative or cerebrovascular disease are more likely to develop delirium [34]. Well-known risk factors of delirium are age, previous dementia history, disease severity during ICU stay, which had shown similar tendencies in our study (Appendix A) [10]. Furthermore, patients with cerebrovascular diseases are at a higher risk of stroke recurrence during treatment in the ICU. In our study, two of three patients who had acute stroke during ICU stay had a history of cerebrovascular disease (data not shown in the tables).

The APACHE II and SOFA scores are useful tools in predicting the clinical outcomes of critically ill patients, and Zhou et al. showed that a higher SOFA score at admission was a risk factor for death in adult patients with COVID-19 [35,36]. Our findings showed that the APACHE II or SOFA score was an independent factor of ICU mortality in COVID-19 patients admitted to the ICU.

This study included all patients from the opening to the closing of the ICU in a government-designated hospital, and we were able to obtain the clinical outcomes of all patients, which are both the strengths of this study. Nevertheless, this study has several limitations. Careful attention should be paid to the interpretation of the results as this is a retrospective study with a small number of patients. Functional status prior to ICU admission such as pre-modified Rankin scale score, which could affect clinical outcome, was not reflected in the analysis because of the limitation of a retrospective study. In addition, we were unable to ascertain causality. New-onset neurological complications could be correlated with other medical complications and ICU death, directly or indirectly; however, these correlations could not confirm causation. Some of the neurological complications could have also been underdiagnosed, such as non-convulsive seizures and unexplained encephalopathy, which are relatively common in ICU patients with sepsis [37]. Our study did not implement an EEG for all ICU patients, only those who developed seizures clinically or had unexplained mental changes. In addition, only stable patients with vital signs underwent MRI. A recent study conducted brain MRIs on 13 patients with unexplained encephalopathic features and found two cases of acute and one case of subacute cerebral infarction; however, neither of the cases had any focal symptoms of a stroke [7].

## 5. Conclusions

In this study, we observed that more than one-third of the patients who were treated in the integrated ICU for COVID-19 infection already had neurological comorbidities. Furthermore, new-onset neurological complications occurred in 44.2% of the patients during ICU stay. Because preexisting neurological comorbidities and new-onset neurological complications during ICU stay have significant effects on the treatment and prognosis of ICU patients, both directly and indirectly, clinical awareness about neurological problems is thought to be important for the treatment of patients hospitalized in the ICU due to COVID-19. Therefore, additional attention to neurological conditions is pertinent to treating COVID-19. This can be accomplished by including neurointensivists in frontline treatment teams.

## Figures and Tables

**Figure 1 jcm-10-02281-f001:**
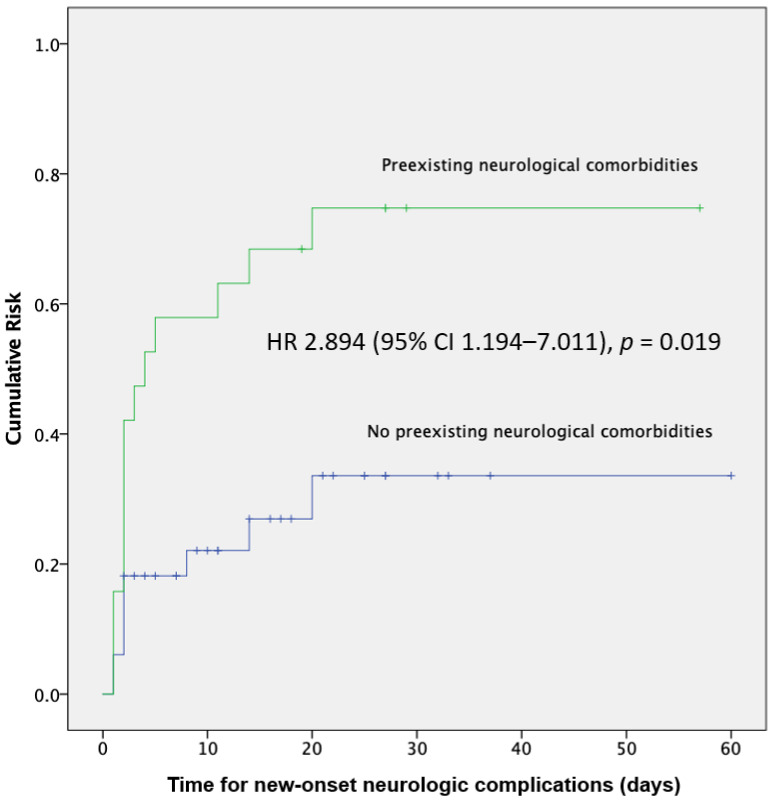
Kaplan–Meier estimates of the cumulative risk of new-onset neurological complications.

**Table 1 jcm-10-02281-t001:** Baseline characteristics of 52 intensive care unit patients infected with COVID-19.

	All ICU Patients N = 52	Preexisting Neurological Comorbidities N = 19 (36.5%)	No Preexisting Neurological Comorbidities N = 33 (63.5%)	*p* Value	New-Onset Neurological Complications N = 23 (44.2%)	No New-Onset Neurological Complications N = 29 (45.8%)	*p* Value
Characteristics							
Age, median (IQR), year	73 (57–81)	79 (59–84)	68 (56–78)	0.114 ^†^	79 (63–85)	68 (54–77)	0.054 ^†^
Sex, male (%)	27 (51.9)	14 (73.7)	13 (39.4)	0.017 ^∫^	13 (56.5)	14 (48.3)	0.755 ^∫^
Underlying medical conditions (%)							
Hypertension	29 (55.8)	10 (52.6)	19 (57.6)	0.730 ^∫^	14 (60.9)	15 (51.7)	0.705 ^∫^
Diabetes mellitus	24 (46.2)	7 (36.8)	17 (51.5)	0.307 ^∫^	9 (39.1)	15 (51.7)	0.532 ^∫^
Dyslipidemia	5 (9.5)	2 (10.5)	3 (9.1)	1 *	4 (17.4)	1 (3.4)	0.222 *
Cardiac disease	9 (17.3)	5 (26.3)	4 (12.1)	0.260 *	6 (26.1)	3 (10.3)	0.262 *
Malignancy	6 (11.5)	2 (10.5)	4 (12.1)	1 *	4 (17.4)	2 (6.9)	0.460 *
Chronic liver disease	2 (3.8)	1 (5.3)	1 (3)	1 *	2 (8.7)	0 (0.0)	0.372 *
Chronic kidney disease	6 (11.5)	4 (21.1)	2 (6.1)	0.175 *	5 (21.7)	1 (3.4)	0.107 *
Scoring system, median (IQR)							
SOFA score	4 (2–6)	4 (2–7)	3 (2–5)	0.343 ^†^	5 (2–7)	2 (1–5)	0.044 ^†^
APACHE II score	11 (7–21)	12 (9–21)	10 (7–20)	0.232 ^†^	12 (9–22)	9 (7–19)	0.040 ^†^
Signs and symptoms (%)	39 (75.0)	13 (68.4)	26 (78.8)	0.406 ^∫^	15 (65.2)	24 (82.8)	0.147 ^∫^
Fever	35 (67.3)	12 (63.2)	23 (69.7)	0.628 ^∫^	13 (56.5)	22 (75.9)	0.238 ^∫^
Chilling	10 (19.2)	2 (10.5)	8 (24.2)	0.293 *	4 (17.4)	6 (20.7)	1 *
Cough	22 (42.3)	5 (26.3)	17 (51.5)	0.077 ^∫^	7 (30.4)	15 (51.7)	0.207 ^∫^
Sputum production	15 (28.8)	6 (31.6)	9 (27.3)	0.741 ^∫^	5 (21.7)	10 (34.5)	0.484 ^∫^
Rhinorrhea	6 (11.5)	2 (10.5)	4 (12.1)	1 *	2 (8.7)	4 (13.8)	0.893 *
Sore throat	9 (17.3)	0 (0)	9 (27.3)	0.018 *	3 (13.0)	6 (20.7)	0.723 *
Myalgia or fatigue	20 (38.5)	2 (10.5)	18 (54.5)	0.002 ^∫^	6 (26.1)	14 (48.3)	0.178 ^∫^
Headache	12 (23.1)	1 (5.3)	11 (33.3)	0.037 *	6 (26.1)	6 (20.7)	0.646 ^∫^
Diarrhea	10 (19.2)	3 (15.8)	7 (21.2)	0.729 *	4 (17.4)	6 (20.7)	1.000 *
Dyspnea	21 (40.4)	5 (26.3)	16 (48.5)	0.117 ^∫^	8 (34.8)	13 (44.8)	0.654 ^∫^
Chest pain	1 (1.9)	1 (5.3)	0 (0)	0.365 *	1 (4.3)	0 (0)	0.907 *

COVID-19, coronavirus disease 2019; IQR, interquartile range; SOFA, Sequential Organ Failure Assessment; APACHE, Acute Physiology and Chronic Health Evaluation. Data are median (interquartile range), n (%), or n/N (%), where N is the total number of patients with available data. *p* values were obtained using χ^2^ test ^∫^, Fisher’ exact test *, or Mann–Whitney U test ^†^.

**Table 2 jcm-10-02281-t002:** Preexisting neurological comorbidities and new-onset neurological complications of 52 intensive care unit patients infected with COVID-19.

	All ICU Patients N = 52	Preexisting Neurological Comorbidities N = 19 (36.5%)	No Preexisting Neurological Comorbidities N = 33 (63.5%)	*p* Value	New-Onset Neurological Complications N = 23 (44.2%)	No New-Onset Neurological Complications N = 29 (45.8%)	*p* Value
Characteristics							
Preexisting neurologic comorbidities (%)	19 (36.5)	19 (100)			14 (60.9)	5 (17.2)	0.001 ^∫^
Neurodegenerative disease	7 (13.5)	7 (36.8)			5 (21.7)	2 (6.9)	0.219 *
Alzheimer disease and other dementias	4 (7.7)	4 (21.1)			4 (17.4)	0 (0)	0.033 *
PD and PD-related disorders	2 (3.8)	2 (10.5)			1 (4.3)	1 (3.4)	1.000 *
Motor neuron disease	2 (3.8)	2 (10.5)			1 (4.3)	1 (3.4)	1.000 *
Cerebrovascular disease	7 (13.5)	7 (36.8)			5 (21.7)	2 (6.9)	0.219 *
Other neurologic disease	5 (9.6)	5 (26.3)			4 (17.4)	1 (3.4)	0.157 *
Complications (%)							
Neurological complications	23 (44.2)	14 (73.7)	9 (27.3)	0.001 ^∫^	23 (100)		
Seizure	3 (5.8)	2 (10.5)	1 (3.0)	0.546 *	3 (13.0)		
Acute stroke	3 (5.8)	3 (15.8)	0 (0)	0.044 *	3 (13.0)		
Delirium	20 (38.5)	13 (68.4)	7 (21.2)	0.001 ^∫^	20 (87.0)		
Acute respiratory distress syndrome	21 (40.4)	9 (47.4)	12 (36.4)	0.436 ^∫^	14 (60.9)	7 (24.1)	0.017 ^∫^
Acute kidney injury	13 (25)	6 (31.6)	7 (21.2)	0.510 *	7 (30.4)	6 (20.7)	0.629 ^∫^
Septic shock	22 (42.3)	10 (52.6)	12 (36.4)	0.253 ^∫^	14 (60.9)	8 (27.6)	0.033 ^∫^

COVID-19, coronavirus disease 2019; IQR, interquartile range; PD, Parkinson’s disease. Data are median (interquartile range), n (%), or n/N (%), where N is the total number of patients with available data. *p* values were obtained using χ^2 ∫^ or Fisher’ exact test*.

**Table 3 jcm-10-02281-t003:** Treatment and prognosis of 52 intensive care unit patients infected with COVID-19.

	All ICU Patients N = 52	Preexisting Neurological Comorbidities N = 19 (36.5%)	No Preexisting Neurological Comorbidities N = 33 (63.5%)	*p* Value	New-Onset Neurological Complications N = 23 (44.2%)	No New-Onset Neurological Complications N = 29 (45.8%)	*p* Value
Treatments (%)							
Mechanical ventilation	26 (50)	11 (57.9)	15 (45.5)	0.388 ^∫^	16 (69.6)	10 (34.5)	0.026 ^∫^
non-invasive facial mask	9 (17.3)	3 (15.8)	6 (18.2)	1 *	4 (17.4)	5 (17.2)	1 *
invasive	23 (44.2)	10 (52.6)	13 (39.4)	0.355 ^∫^	15 (65.2)	8 (27.6)	0.015 ^∫^
Extracorporeal membrane oxygenation	5 (9.6)	3 (15.8)	2 (6.1)	0.342 *	4 (17.4)	1 (3.4)	0.222 *
Continuous renal replacement therapy	8 (15.4)	5 (26.3)	3 (9.1)	0.124 *	6 (26.1)	2 (6.9)	0.129 *
Antibiotic treatment	49 (94.2)	18 (94.7)	31 (93.9)	1 *	23 (100)	26 (89.7)	0.245 *
Antiviral treatment	49 (94.2)	17 (89.5)	32 (97)	0.546 *	23 (100)	26 (89.7)	0.245 *
Glucocorticoids	39 (75)	14 (73.7)	25 (75.8)	1 *	20 (87.0)	19 (65.5)	0.147 ^∫^
Intravenous immunoglobulin therapy	1 (1.9)	0 (0)	1 (3)	1 *	0 (0)	1 (3.4)	1 *
Tracheostomy	9 (17.3)	7 (36.8)	2 (6.1)	0.008 *	6 (26.1)	3 (10.3)	0.262 *
Treatment Timeline, median (IQR), days—n/N.							
length of ICU stays	22 (9–32)	27 (18–32)	16 (7–30)	0.048 ^†^	26 (8–38)	19 (10–27)	0.361 ^†^
Invasive ventilator days, 23/52	18 (3–29)	23 (18–37)10/19	5 (2–20)13/33	0.013 ^†^	18 (5–30)15/23	7 (2–28)8/29	0.437 ^†^
Onset-to-ICU admission, 39/52	8 (6–11)	8 (7–10)13/19	9 (5–11)26/33	0.419 ^†^	8 (7–11)15/23	8 (5–11)24/29	0.674 ^†^
Prognosis—n/N. (%)							
Transfer out of ICU	33/52 (63.5)	10/19 (52.6)	23/33 (69.7)	0.218 ^∫^	10/23 (43.5)	23/29 (79.3)	0.008 ^∫^
ICU death	16/48 (33.3)	9/19 (47.4)	10/33 (30.3)		13/23 (56.5)	6/29 (20.7)	

COVID-19, coronavirus disease 2019; ICU, intensive care unit; IQR, interquartile range. Data are median (interquartile range), n (%), or n/N (%), where N is the total number of patients with available data. *p* values were obtained using χ^2^ test ^∫^, Fisher’ exact test *, or Mann–Whitney U test ^†^.

**Table 4 jcm-10-02281-t004:** Independent predictors associated with ICU mortality in the multiple logistic regression analysis using APACHE II score.

**Model 1**	**OR**	**95% C.I.**	***p*-Value**
Age	1.01	0.95–1.07	0.81
Male	4.15	0.79–21.84	0.09
APACHE II score	1.20	1.06–1.36	0.003
New-onset neurologic complications	6.18	1.16–32.90	0.03
Nagelkerke *R*^2^ = 0.576 Hosmer–Lemeshow Chi-square test = 7.955, *df* = 8, *p*-value = 0.438			
**Model 2**	**OR**	**95% C.I.**	***p*-Value**
Age	1.01	0.95–1.08	0.73
Male	5.23	0.82–33.23	0.08
APACHE II score	1.21	1.06–1.37	0.004
Preexisting neurological comorbidities	0.55	0.77–3.88	0.55
New-onset neurologic complications	7.86	1.21–51.32	0.03
Nagelkerke *R*^2^ = 0.581 Hosmer–-Lemeshow Chi-square test = 4.141, *df* = 8, *p*-value = 0.844			

CI, confidence interval; ICU, intensive care unit; OR, odds ratio; APACHE, Acute Physiology and Chronic Health Evaluation.

## Data Availability

The data presented in this study are available on request from the corresponding author.

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
