# Peer review of "Clinical Implications of Neurological Comorbidities and Complications in ICU Patients with COVID-19"

_jcm, 2021, doi:10.3390/jcm10112281_

Round 1
Reviewer 1 Report
The authors conducted a retrospective cohort study to investigate the impact of preexisting neurological comorbidities and new-onset neurological complications on clinical outcomes in critically-ill patients infected with COVID-19.
Neurological comorbidities and neurological complication is a common phenomenon especially in patients with severe covid-19 infection,
yet the their clinical implications remain unknown. The authors meticulously examined the neurological conditions in this patient population. Further large-scale studies are warranted.
The text is well-organized and well-written with logical formatting, and it is of appropriate length.
I have the following comments.
1. Seizure has not been widely reported in patients with COVID-19.
Additional comments on biological mechanism of the link between COVID-19 and seizure would help to understand the nature of COVID-19 infection with neurological complication.
2. Line 323-328
As the authors already mentioned, this study cannot explain a causal relationship between neurological conditions and clinical outcomes. Therefore, unless the conditions including delirium, acute stroke and seizure directly leaded to poor prognosis, the effects of such conditions should be de-emphasized.
Author Response
Response 1: Thank you for giving me a good opinion. I added as below in discussion section.
Seizure has not been widely reported in patients with COVID-19. SARS-CoV-2 infection may cause multi-organ failures, metabolic derangements, and hypoxia, which can lead to clinical or subclinical seizures.[25] Acute cerebrovascular disease is considered a major cause of seizure.
- Lu, L.; Xiong, W.; Liu, D.; Liu, J.; Yang, D.; Li, N.; Mu, J.; Guo, J.; Li, W.; Wang, G.; et al. New Onset Acute Symptomatic Seizure and Risk Factors in Coronavirus Disease 2019: A Retrospective Multicenter Study. Epilepsia 2020, 61, e49–e53, doi:10.1111/epi.16524.
Point 2: Line 323-328
As the authors already mentioned, this study cannot explain a causal relationship between neurological conditions and clinical outcomes. Therefore, unless the conditions including delirium, acute stroke and seizure directly leaded to poor prognosis, the effects of such conditions should be de-emphasized.
Response 2: Thank you for your good point. We made some modifications to what the reviewer mentioned as follows.
ARDS, acute cardiac injury, and septic shock have been reported to occur frequently in ICU patients with COVID-19.[30,31] Neurological complications may be caused by COVID-19-related medical complications. [32][7] Although this study cannot explain a causal relationship, patients who had new-onset neurological complications during ICU care had a higher ICU mortality rate than those that did not. Delirium in the ICU has a negative effect on the outcome of patients. [33,34] Acute stroke or seizure can also directly or indirectly affect a patient’s treatment and prognosis.

Reviewer 2 Report
The article is nicely written. Data analysis was conducted with sound methods. I accept the article with minor suggestion below:
- How did they estimate hazard rate in Cox's regression is not specified. Authors can add a sentence or two in the statistical analysis section. They used a software package but there must be a criterion to decide the hazard functions.
Author Response
Point 1: How did they estimate hazard rate in Cox's regression is not specified. Authors can add a sentence or two in the statistical analysis section. They used a software package but there must be a criterion to decide the hazard functions.
Response 1: Thank you for a wonderful review. We have revised the manuscript, as below.
In the Method section,
Statistical analysis was performed by multiple logistic regression analysis and Cox proportional hazard analysis. Predefined variables (age, sex, and APACHE II score) were adjusted in the statistical analysis because of an overfitting issue.
- Cox propotional hazard model was applied to evaluate the influences on new onset neurological complications according to preexisting neurological comorbidities. Multiple logistic regression analysis was used to evaluate the independent predictors of ICU mortality in ICU patients with COVID-19. Predefined variables (age, sex, and APACHE II score) were adjusted in the both statistical analysis models because of an overfitting issue.
Reviewer 3 Report
The manuscript by Park et al. investigates the role of preexisting neurological comorbidities and the onset of new neurological complications in patients with severe COVID-19 that were treated in their intensive care unit. The Authors concluded that the onset of new neurological comorbidity is an independent factor for COVID-19-associated mortality. My comments are specifically towards the need to clarify the differences between the new onset of neurological complications when compared to worsening neurological status in already pre-existing comorbidity.
- That being said, it is important to compare the neurological complications between patients without pre-existing neurological comorbidities versus the ones that already had neurological diseases (9 versus 14 patients). Were there any differences in which neurological complications were occurring? And differences in their COVID-19 outcomes? For example, someone could hypothesize that patients with established AD may mainly develop delirium when compared to patients without neurological comorbidities who will develop an ischemic complications.
- In my opinion, there are actually three groups in this study population 1) patients without pre-existing neurological comorbidities and no neurological complications, 2) patients with pre-existing neurological comorbidities and no neurological complications, and 3) patients that had both pre-existing and new-onset. Kaplan-Meier curves regarding mortality, mechanical ventilation between these 3 groups would be of particular interest.
- Please indicate which test was used for which comparison in the Tables of the manuscript. Also please indicate what test/procedure was used to determine the data distribution (Kolmogorov-Smirnov vs. Shapiro-Wilks vs. Q-Q plots and histograms). Given that all measures are shown in medians (as non-parametric data) the comparison with t-test cannot be replicated.
- What is the reason for the two logistic regression models in Table 4? I believe the only second model with gender as a predictor should remain.
- An important practical and clinical question is whether the presence of pre-existing neurological diseases a factor of outcomes rather in addition to new onset of neurological complications? The addition of this variable in a supplementary model would answer the question.
- Lastly, what percentage of variance in the COVID-19 outcome can contribute to these findings? Providing R2 for the models would help.
